# Effects of Truffle Inoculation on Root Physiology and Mycorrhizosphere Microbial Communities of *Carya illinoinensis* Seedlings

Haoyu Chen [1,2], Jiawei Wu [1,2], Junping Liu [1,2], Pengpeng Tan [1,2], Kaikai Zhu [1,2] and Fangren Peng [1,2,*]

1   Co-Innovation Center for Sustainable Forestry in Southern China, Nanjing Forestry University, Nanjing 210037, China; 13515100963@163.com (H.C.); xichuan@njfu.edu.cn (J.W.); ningmengzhiyuan12@163.com (J.L.); tanpengpeng2002@163.com (P.T.); kkzhu@njfu.edu.cn (K.Z.)
2   College of Forestry, Nanjing Forestry University, Nanjing 210037, China
*   Correspondence: frpeng@njfu.edu.cn

**Abstract:** Although they are a valuable edible ectomycorrhizal fungus, truffles (*Tuber* spp.) nevertheless face significant difficulties in the development of their scale. As a type of high economic value nut-like economic forest tree, the pecan (*Carya illinoinensis*) serves as a natural host for truffles. However, the technology for mycorrhizal synthesis in pecan has not yet been developed, and it is still unclear how certain microbes affect this process. In this study, we inoculated the pecan root system with a suspension of truffle spores and investigated the growth physiology of pecan seedlings with various infestation levels, as well as changes in the soil's physicochemical properties and the makeup of the microbial community at the root level. The findings showed that tuber inoculation significantly increased the peroxidase activity of the pecan root system, significantly decreased the pH, and effective phosphorus content of mycorrhizosphere soil, while increasing the nitrate nitrogen content, and significantly increased the abundance and diversity of the mycorrhizosphere soil fungal community. Different groups of fungal and bacterial markers were formed in the mycorrhizosphere of pecan seedlings at different levels of infestation. In the highly infested group, *Rozellomycota* and *lasiosphaeriaceae* were the difference marker fungi, and *Xanthobacteraceae*, *Rhizobiaceae* as well as *Streptococcaceae* were the difference marker bacteria. In the low-infestation group, *sphaerosporella* was differential marker fungi, and *Bacillus* and *Tumebacillus* were differential marker bacteria. The fungal marker flora of the control group consisted of *Chaetomium* and *Gilmaniella*. *Pseudomonas* was the marker bacterial community. Additionally, these fungi included *Collarina* and *Rozellomycota*, and several bacteria from the genera *Pseudomonas*, *Gemmatimonas*, and others showed highly significant relationships with changes in soil pH, effective phosphorus, and nitrate nitrogen. In conclusion, pecan–truffle mycorrhizal seedlings have the potential to create the ideal microbial community structure needed for mycorrhizal growth, and these microorganisms have the potential to significantly alter the pH, effective phosphorus content, and nitrate nitrogen concentration of the mycorrhizosphere soil. Our results contribute to the understanding of how the mycorrhizosphere microbial community evolves when exogenous mycorrhizal fungi infest host plants and can offer some theoretical guidelines for growing pecan–truffle mycorrhizal seedlings.

**Keywords:** pecan; truffles; ectomycorrhizae; high-throughput sequencing; mycorrhizosphere microbiome; soil physicochemical properties





## 1. Introduction

*Tuber* spp. refers to a group of large edible fungi, the majority of which are members of the genus *Tuber* (*Tuberaceae*) within the order Discomycetes of the phylum Ascomycetes. These fungi produce ascomycetes either above or below ground and their fruiting bodies are known as truffles [1]. Although truffles have received a lot of attention from researchers in the field of forest fungal complex management due to their great culinary

and medicinal value, people's knowledge of truffles is still quite restricted and there are still many unanswered questions regarding the spatial and temporal distribution of the mycelial mating type and the process of ascomycete fruit formation [2,3], which adds to the uncertainty in truffle production and is one of the factors contributing to their scarcity. Although some details of the life history of truffles remain to be elucidated, their production under semi-artificially controlled conditions is still necessary. Currently, artificial inoculation of spore suspensions into plant root systems is the primary method used to develop truffles [4]. Truffles belong to the group of ectomycorrhizal fungi (EMF), which can form ectomycorrhizas with several plants of *Juglandaceae*, *Salicaceae*, *Fagaceae*, and *Pinaceae*, among others [5–7]. This relationship is often considered as a reciprocal symbiosis [8]. Through the massive mycorrhizae and their mycelial system, EMF multiplies the host plant's root system's absorption area by hundreds or thousands of times [9]. Additionally, EMF increases trees' capacity to absorb water and mineral nutrients by promoting biological nitrogen fixation and hastening the weathering and release of soil constituents like phosphorus, potassium, and calcium [10,11]; on the other hand, the plant provides the fungus with organic substances such as carbohydrates to satisfy the needs of its growth and development [12]. Pecan (*Carya illinoinensis*), as an important economic tree species and a natural host for truffles [13], has a large potential for industrial cultivation with truffles. In 2016, Marozzi et al. [14] succeeded in realizing the artificial inoculation of *Tuber melanosporum* with *Tuber brumale* on pecan, while in the actual process of artificial cultivation of truffles, there is always suffered from low mycorrhizal infestation rates.

Numerous studies have shown that the infestation of host plants by ectomycorrhizal fungi is influenced by a variety of factors, including abiotic factors such as temperature, humidity, pH, geography, and soil nutrient status [15], as well as biotic factors such as truffle-feeding animals, chemosensory plants, and microbials in the soil [16,17]. A growing number of researchers have concentrated on the impact of soil microorganisms among these numerous effects. Following the concept of Mycorrhization helper bacteria (MHB) [18], many bacteria have been shown to promote the synthesis of exogenous mycorrhizae. For example, *Bacillus cereus* has been shown to significantly increase the activity of EMF, which increases the rate of infestation [19]. *Pseudomonas fluorescens* has also been demonstrated to promote the mycorrhization levels of EMF [20]; Sabella et al. [21] found that *Arthrinium phaeospermum* was able to promote the formation of ectomycorrhizae of *Tuber borchii* by shortening the primary root system of the host plant and enhancing growth in secondary roots, exploring the possibility of the existence of mycorrhizal helper fungi. These beneficial interactions or antagonistic interactions between mycorrhizal fungi and soil microorganisms directly or indirectly affect the composition and abundance of the host mycorrhizosphere microbial community, regulating the composition of the soil microflora and maintaining it in a dynamic ecosystem equilibrium [22]. Mycorrhizal fungi play a vital role in this process as indicators, participants, and facilitators [23]. However, little research has been carried out on the modifications that a tuber fungus infestation causes in the mycorrhizosphere microbial community of the host plant. Thus, the interaction between microorganisms and mycorrhizal fungi and the structural properties of the microbial community of pecan–truffles mycorrhizosphere under artificial inoculation remain unclear. It has been demonstrated that plants have the capacity to select particular microorganisms to sculpt the mycorrhizosphere microbial community [24]. It is conceivable that plants may enlist particular microorganisms to encourage mycorrhizal development during interactions between mycorrhizal fungus and host plants.

Hence, the objectives of our study were to reveal (i) differences in microbial diversity and community structure due to truffle infestation; (ii) correlations between soil chemical properties and microbial communities; and (iii) the relationship between truffle infestation and root physiology. In this study, we used high-throughput sequencing techniques (16S and ITS) to examine the mycorrhizosphere microbial community of pecan mycorrhizae following tuber inoculation. We also examined changes in the structural characteristics of the pecan mycorrhizosphere microbial community after tuber inoculation, while also

paying attention to changes in the mycorrhizosphere soil nutrients of the seedling and the physiology of the root system. The study's findings offer some theoretical points of reference for explaining the microbial processes involved in pecan mycorrhizal synthesis and the propagation of pecan–truffle mycorrhizal seedlings.

## 2. Materials and Methods

### 2.1. Seedling and Tuber Inoculation of Pecan

In November 2021, pecan seeds of Pawnee were obtained from the pecan test base of Nanjing Forestry University in Jurong, Jiangsu Province. The seeds were steeped in water for 3–5 d after being sterilized with a 0.2% potassium permanganate solution. The sterilized seeds were evenly scattered on top of the sand bed so that they did not overlap, covered with 5 cm of clean river sand, and stored in sand at low temperatures for 2 months. The soaked seeds were germinated by sand bed stratification, which was set up on a well-drained, flat ground. Afterwards, the seeds were rinsed, surface sterilized, and placed in a nursery with a perlite, vermiculite, and peat soil combination in an equal volume ratio for germination.

In May 2022, seedlings with robust root systems, a high number of lateral roots, and consistent growth conditions were selected for inoculation. Organic cultivation substrate (Jiangsu Xingnong Substrate Technology Co., Ltd., Zhenjiang, China) was autoclaved and used for transplanting seedlings after inoculation. The background pH of the substrate was 6.49, and the contents of total nitrogen (TN), total phosphorus (TP), and total potassium (TK) were 5.83 g/kg, 1.25 g/kg, and 8.17 g/kg, respectively, while the content of soil organic carbon (SOC) was 112.90 g/kg, and the contents of quick-acting phosphorus (AP) and effective potassium (AK) were 257.29 mg/kg and 523.94 mg/kg, respectively. Containers were made from cylindrical non-woven bags with a diameter of 25 cm and a height of 30 cm. The test truffles were purchased from Yunnan Hanlu Foods Co., Ltd., Yunnan, China and underwent morphological testing and molecular characterization. Blast comparison of the representative sequences revealed that the homology with Tuber indicum was as high as 99.60%.

The spore suspension method was used to inoculate the truffles, the fruit body of truffle was ground into a powder using a pulverizer, and 2.0 g of the ground truffle powder was weighed into a tiny beaker. To this was added 20 mL of pure water to create the microbial inoculum. After adequately root-breaking the seedlings, the microbial inoculum was applied to the root system and left on for two to three minutes before being poured around the root system and the substrate being filled. The number of truffles used to inoculate each seedling was 2 g, and the spore concentration was about $6.19 \times 10^6$/g as measured by the hemocyte counting plate. After inoculation, the test seedlings were planted in Jurong City, Jiangsu Province, China's pecan experimental base of Nanjing Forestry University. Test seedlings were uniformly managed and watered according to the soil's dryness and wetness. Every month, 3–5 seedlings were sampled and tested, and the seedlings were harvested in the sixth month after a significant number of mycorrhizal roots were found. The test seedlings, consisting of the control and inoculated groups, each had 17 plants (surviving). There were 34 plants in total.

### 2.2. Sample Collection

All test seedlings were harvested six months after inoculation, their roots were cut off, the large soil samples were shaken off, and the soil [25] adhering to 0–3 mm around the fine roots was collected with a small brush and stored as mycorrhizosphere soil samples in a refrigerator at −80 °C to determine the microbial indexes as well as the soil's physicochemical properties. To determine root physiological indicators, seedling roots were cleaned and put in sterile bags. Despite the fact that we artificially inoculated the pecan with truffles, in the wild, other uncontrollable circumstances may prevent the truffles from becoming the majority population in the mycorrhizosphere soil of the test seedlings. To better show the effect of the truffle population on the mycorrhizosphere of pecan mycorrhizal fungus, based

on the results of the infestation results (Supplementary Table S1), the three seedlings with the highest infestation rate were set as the high-infestation group and the three seedlings with the lowest infestation rate were set as the low-infestation group. The three groups were divided into the high-infestation group (HI), the low-infestation group (LI), and the blank control group (CK) for the determination of the indexes.

*2.3. Colonization Detection and Infestation Rate Statistics of Tubercle Bacteria*

2.3.1. Morphological Detection of ECM

In September 2022, mycorrhizal formation was detected for the first time, but the infestation rate was low, and all the test seedlings were harvested in November to determine the colonization of truffles in pecan from the observation of root morphology, anatomical structure, and molecular identification. Under the illumination of a daylight-type light source, the mycorrhizae were seen with a stereomicroscope, and macroscopic aspects including color, form, branching, and the degree of mycelial looseness of the mycorrhizae were described in accordance with the method of Agerer [26]. Using a micrographic method, the morphological traits of mycorrhizae were documented. A mycorrhizal section was produced and Safranine and Fast Green double dyeing were used. Epiphytic hyphae and Hartig net (the structure through which EMF shares nutrients with the host plant, providing clear evidence of mycorrhizal development) should be observed. In addition, a section of mycorrhizal root was placed in a glass dish containing water, and the mantle was gently peeled off with tweezers and observed under a light microscope under a stereomicroscope. The smoothness of the surface of the mycelium, the diameter, length, sparseness and arrangement of the mycelium, and the presence or absence of epitaxial hyphae and cystidium were characterized.

2.3.2. Molecular Identification of Mycorrhizal Fungi

ITS1 and ITS4 universal primers for fungi were chosen for sequencing after the DNA of fresh mycorrhizal fungi from pecan tuber was extracted (Shanghai Meiji Biomedical Technology Co., Ltd., Shanghai, China). Following sequencing, NCBI's Blast comparison was used to identify the species of mycorrhizal fungi with the most comparable sequences that scored highly.

2.3.3. Calculation of Infestation Rate

After mycorrhizal seedlings were harvested, the roots were grabbed and the substrate covering the root system shaken off. The root system was then slowly rinsed under running water, submerged in a large beaker of water, and gently agitated to wash away any impurities adhering to the root system's surface. The excess water was then sucked off with absorbent filter paper. One branch root from the upper, middle, and lower portions of the root system was removed for mycorrhizal counting, and the mycorrhizae were counted under a body microscope [27]. The infestation rate $R$ was calculated in accordance with the estimated infestation rate (Supplementary Table S1), and the inoculated test seedlings were grouped.

$$R = \left( \frac{Total\ number\ of\ mycorrhizae\ on\ 3\ feeder\ roots}{Total\ number\ of\ lateral\ roots\ on\ 3\ feeder\ roots} \right) \tag{1}$$

$R$ is the infestation rate.

*2.4. Determination of Sample Indicators*

2.4.1. Steps for the Determination of Microbial Indicators

Deoxyribonucleic acid (DNA) was extracted from mycorrhizosphere soil samples of three replicates from each treatment group using a FastDNA Spin Kit for Soil (MP Bio) kit in accordance with the manufacturer's instructions. The isolated DNA samples were kept at $-20\ ^\circ C$ and their quality was evaluated using gel electrophoresis.

The V1–V3 region of the bacterial 16S gene was amplified using the bacterial primers 27F (5'-AGAGTTTGATCCTGGCTCAG-3') and 533R (5'-TTACCGCGGCTGCTGGGCAC-3'); the ITS1 region of the fungal ITS gene was amplified using the universal primers ITS1F (5'-CTTGGTCATTTAGAGGAAGTAA-3') and ITS2R (5'-GCTGCGTTCTTCATCGATGC-3'). PCR amplification system: 27F/533R PCR was performed using TransGen AP221-02: TransStart Fastpfu DNA Polymerase. TaKaRa rTaq DNA Polymerase was used for ITS1F/ITS2R; the PCR instrument was ABI GeneAmp&reg; Model 9700; 3 replicates were used for each sample, and the PCR products of the same sample were mixed and detected by 2% agarose gel electrophoresis, and the gels were cut by using the AxyPrepDNA Gel Recovery Kit (AXYGEN). PCR products were detected by 2% agarose electrophoresis. Referring to the preliminary quantitative results of electrophoresis, the PCR products were quantified by QuantiFluor™ -ST Blue Fluorescence Quantification System (Promega), after which the PCR products were mixed according to the corresponding proportion of each sample according to the sequencing amount required. Illumina library construction was performed with the TruSeqTM DNA Sample Prep Kit, and the libraries were sequenced using an Illumina MiSeq sequencer (Personalbio, Shanghai, China). PE reads obtained from Illumina sequencing were firstly spliced according to the overlap relationship, and the sequence quality was also analyzed. The PE reads obtained from Illumina sequencing were firstly spliced according to the overlap relationship, and at the same time, the quality of the sequences was quality controlled and filtered, and the samples were differentiated for OTU clustering analysis and species taxonomic analysis. Based on the results of OTU clustering analysis, a variety of diversity indices can be analyzed for OTUs, as well as the detection of the sequencing depth; based on the taxonomic information, the statistical analysis of community structure can be carried out at each taxonomic level. On the basis of the above analysis, a series of in-depth statistical and visualization analyses, such as multivariate analysis and significance of difference test, can be carried out on the community composition and phylogenetic information of multiple samples.

### 2.4.2. Measurement Steps of Soil Physicochemical Indexes

Soil physicochemical indicators include soil pH, ammonium nitrogen ($NH_4^+$-N), nitrate nitrogen ($NO_3^-$-N), quick-acting phosphorus (AP), quick-acting potassium (AK), and exchangeable calcium ($Ca^{2+}$) were measured. The soil samples were leached with $CO_2$-free distilled water with a water-soil mass ratio of 2.5:1, and the soil pH was determined by a pH meter. The soil samples were extracted with 2 mol/L potassium chloride (KCL) solution, and after extraction, the colorimetric method using indophenol blue and dual wavelength colorimetric method (wavelengths of 225 nm and 275 nm, respectively; UV spectrophotometer model Shimadzu UVmini-1240, Agilent Technologies, Kyoto, Japan) was used.

Included in this are the measurements of soil pH, exchangeable calcium ($Ca^{2+}$), quick-acting phosphorus (AP), quick-acting potassium (AK), ammonium nitrogen ($NH_4^+$-N), and nitrate nitrogen ($NO_3^-$-N). The soil pH was measured using a pH meter after the soil samples were leached with $CO^2$-free distilled water at a 2.5:1 water-to-soil mass ratio. The soil samples were extracted with a 2 mol/L potassium chloride (KCL) solution, and then the indophenol blue and dual wavelength colorimetric methods (wavelengths of 225 nm and 275 nm, respectively) were used with a UV spectrophotometer model (Shimadzu UVmini-1240, Agilent Technologies, Kyoto, Japan) for the determination of $NH_4^+$-N and $NO_3^-$-N contents; flame photometric method (BWB XP, BWB Technologies, Newbury, UK) for the determination of AK and $Ca^{2+}$ contents; and sodium bicarbonate leaching and molybdenum antimony colorimetric method for the determination of AP contents [28,29].

### 2.4.3. Steps for the Determination of Physiological Indices of the Root System

Growth indices such as seedling height, ground diameter, root dry weight, and crown dry weight were determined and seedling index was calculated [30]. Measurements of

physiological indicators involved evaluating the plant's root vigor, peroxidase activity (POD), and superoxide dismutase (SOD) activity. Test seedlings' root vigor was assessed using the triphenyl tetrazolium chloride (TTC) reduction method [31], superoxide dismutase (SOD) was assessed using the guaiacol colorimetric method [32], and plant peroxidase (POD) was assessed using the NBT method [33].

$$Seeding\ index\ \left( \frac{stem\ diameter}{shoot\ height} + \frac{root\ dry\ weight}{crown\ dry\ weight} \right) \times total\ weight \qquad (2)$$

### 2.5. Data Analysis and Visualization

Data on physiological indicators of seedling growth, soil properties, and $\alpha$-diversity (OTU number, Simpson and Shannon indices) were statistically analyzed using SPSS version 26.0 (IBM, Armonk, NY, USA), and one-way analysis of variance (ANOVA) using Duncan's Multiple Polar Difference Test was employed. $p < 0.05$ was considered statistically significant.

Operational taxonomic unit (OTU) clustering was performed using UPARSE software (version 7.1), and sequences with ≥97% similarity were assigned to the same OTU. OTUs were classified using the Silva (bacterial database) and Unite (fungal database) databases. In order to examine the differences between dominant genera in various samples as well as the phylogenetic relationships of various OTUs, multiple sequence comparisons were carried out using MAFFT software (Version 7.2) [34]. The Shannon and Simpson diversity indices and the number of observed OTUs were used to assess the diversity. Principal coordinate analysis (PCoA) and weighted pairwise clustering techniques (using UPGMA to construct a tree structure and Bray–Curtis distance algorithm) were used to achieve hierarchical clustering, and QIIME (V1.9.1) software was used to calculate the Beta diversity distance matrices. The bacteria that varied in the samples were found using linear discriminant analysis (LDA) [35]. Redundancy analysis (RDA) based on the OTU level to analyze the relationship between soil chemical properties and soil microorganisms. Graphing was performed using R (version 3.3.1). The data were visualized on the cloud platform of Shanghai Meiji Biomedical Technology Co.

## 3. Results

### 3.1. Morphological Characteristics of Ectomycorrhizae

As shown in Figure 1, the root system of the pecan produced typical ectomycorrhizae after truffle inoculation. The root system was thicker than that of the uninoculated pecan seedlings, with more and shorter forks, the tips of which were frequently enlarged (Figure 1A,C), and the mycelia could be seen to be obviously extended from the roots when viewed under the microscope after being sliced (Figure 1E). Part of the root system can be observed to have a sheath-like structure called a mycorrhizal mantle, which is a black, rubbery-looking layer covering the surface of the roots (Figure 1F). Mycorrhizal mantle was peeled off with tweezers under the microscope and can be observed in the entangled epitaxial hyphae (Figure 1H), along with irregularly shaped epidermis cells (Figure 1G). Uninoculated seedling lateral roots were observed as being slender and relatively few in bifurcation (Figure 1B,D). Safranine and Fast Green double dyeing staining of mycorrhizal and control root cross sections revealed that the control roots' cells were organized more sprawlingly and cleanly, and Hartig net development was not visible (Figure 2). In contrast, the cell arrangement in the cross section of the infected pecan roots was more wrinkled and compact. While the mycelium of EMF invaded the cellular interstices of the pecan root system's epidermal layer, it did not reach the cortex, forming an obvious Hatty's web structure that was consistent with the description of the typical ectomycorrhizal structure of angiosperms [36].

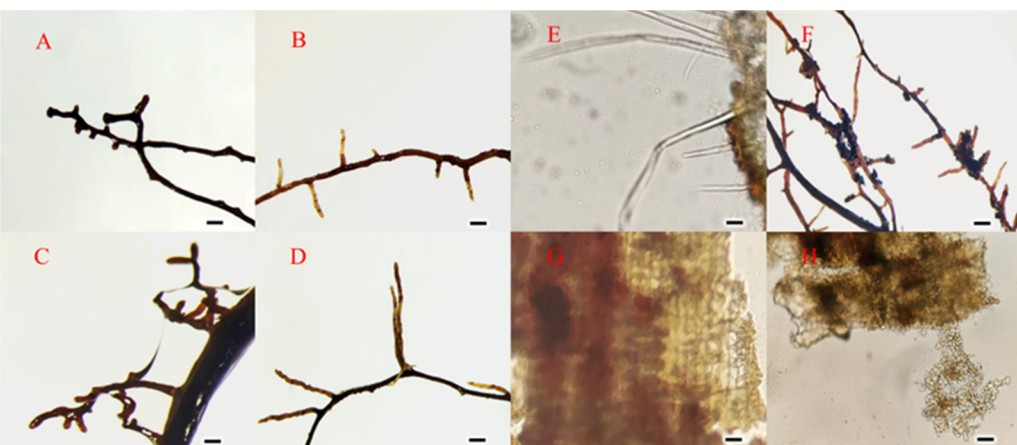

**Figure 1.** Apparent characteristics of *Carya illinoinensis* root with or without Tuber partner. Scale bar = 1 mm (**A**,**C**) Ectomycorrhizae of Tuber, scale bar = 1 mm; (**B**,**D**) *Carya illinoinensis* root without Tuber partner, scale bar = 1 mm; (**E**) epitaxial hyphae observed on hand section of the ectomycorrhizae, scale bar = 100 μm; (**F**,**G**) the mantle in the surface of the ectomycorrhizae, F scale bar = 1 mm, G scale bar = 100 μm; (**H**) entangled hyphae observed within the mantle, scale bar = 100 μm.

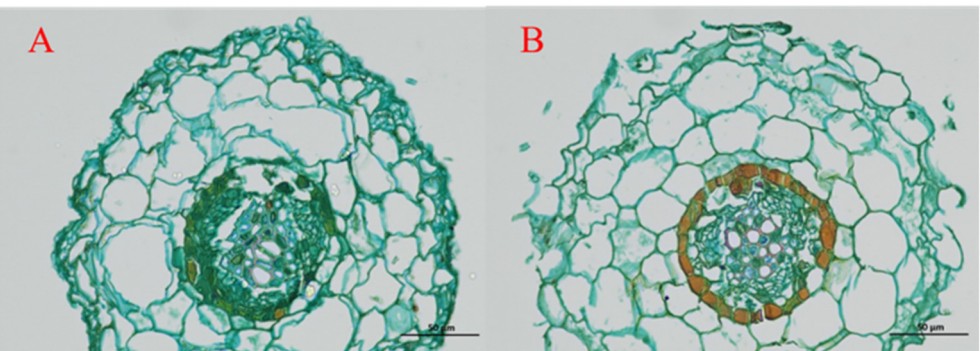

**Figure 2.** Sections of *Carya illinoinensis* root stained with Safranine and Fast Green, scale bar = 50 μm. (**A**) Ectomycorrhizal of Tuber; (**B**) *Carya illinoinensis* roots without Tuber partner.

### 3.2. Physiological Characteristics of Seedling Root System

As shown in Table 1, the average seedling quality index after tuber inoculation ranged from 9.15 to 11.80 at the 6th month, and the root vigor and SOD activity ranged from 2.50 to 4.99 mg TTF/(g·h), 363.28 to 369.70 (U/mg·prot). The three groups of HI, LI, and CK did not differ significantly in the seedling quality index, and the root vigor and root SOD activity were not significantly different either. The average root POD activity of HI was 62.22 (U/mg·prot), which was significantly greater than that of LI and CK.

**Table 1.** Results of multiple comparisons of physiological indices of pecan seedlings with different levels of infestation.

| Group | Infection Rate | Seedling Index | Root Activity mg TTF/(g·h) | SOD Activity (U/mg·prot) | POD Activity (U/mg·prot) |
|---|---|---|---|---|---|
| CK | 0 | 11.80 ± 5.95 a | 2.50 ± 1.35 a | 363.28 ± 44.73 a | 46.67 ± 11.55 b |
| LI | 3.28% | 9.15 ± 4.24 a | 3.05 ± 0.50 a | 369.70 ± 56.48 a | 37.78 ± 3.85 b |
| HI | 56.25% | 9.86 ± 3.29 a | 4.99 ± 2.61 a | 364.75 ± 71.68 a | 62.22 ± 3.85 a |
| *p*-value | | 0.777 | 0.277 | 0.990 | 0.018 |
| F-value | | 0.263 | 1.600 | 0.010 | 8.455 |

Note: values are mean ± standard deviation ($n = 3$). Values followed by different lowercase letters indicate significant differences ($p < 0.05$) among samples within a line. CK is the uninoculated control group, LI is the low-infestation group, and HI is the high-infestation groupings.

### 3.3. Characterization of Mycorrhizosphere Soil Chemical Properties

As can be seen from Table 2, the mycorrhizosphere soil PH of seedlings showed a significant lowering trend with increasing infestation rate and the mycorrhizosphere soil $NH_4^+$-N content ranged between 9.15 and 11.80 (mg/kg). The $NH_4^+$-N content of LI did not differ significantly from that of CK, whereas HI differed significantly greater than CK. The $NO_3^-$-N content ranged from 0.42 to 5.19 (mg/kg), HI was significantly higher than that of LI, and LI was higher than that of CK; whereas, the AP content ranged from 16.96 to 24.38 (mg/kg), HI was significantly lower than that of LI, and LI was significantly lower than that of CK, and the AK content ranged from 77.4 to 103.59 (mg/kg), and there was no significant difference between AK and LI in CK, but AK content in CK was significantly higher than that in HI. Soil $Ca^{2+}$ content of seedlings in the three groups was not significantly different, and the average content ranged from 977.11 to 1259.58 (mg/kg).

**Table 2.** Results of multiple comparisons of physicochemical indexes of mycorrhizosphere soils of pecan with different levels of infestation.

| Group | PH | $NH_4^+$-N (mg/kg) | $NO_3^-$-N (mg/kg) | AP (mg/kg) | AK (mg/kg) | $Ca^{2+}$ (mg/kg) |
|---|---|---|---|---|---|---|
| CK | 6.33 ± 0.01 a | 21.98 ± 5.30 b | 0.42 ± 0.05 c | 24.38 ± 1.25 a | 103.59 ± 11.29 a | 1026.16 ± 54.19 a |
| LI | 6.27 ± 0.06 b | 26.40 ± 3.82 ab | 1.53 ± 0.26 b | 20.55 ± 0.71 b | 99.80 ± 12.40 ab | 977.11 ± 68.44 a |
| HI | 6.08 ± 0.03 c | 32.73 ± 4.73 a | 5.19 ± 0.27 a | 16.96 ± 0.73 c | 77.40 ± 11.77 b | 1259.58 ± 226.07 a |
| *p*-value | <0.0001 | 0.077 | <0.0001 | <0.0001 | 0.069 | 0.099 |
| F-value | 141.848 | 4.044 | 383.220 | 47.458 | 4.300 | 3.491 |

Note: values are mean ± standard deviation (*n* = 3). Values followed by different lowercase letters indicate significant differences (*p* < 0.05) among samples within a line. CK is the uninoculated control group, LI is the low-infestation group, and HI is the high-infestation groupings.

### 3.4. Characterization of Mycorrhizosphere Soil Microbial Communities

A total of 606,259 sequences from ITS high-throughput sequencing were obtained; these sequences were divided into 639 fungal OUTs. As can be seen in Table 3, there was no discernible difference between the three treatments' OTU counts, which varied from 256 to 308. There was no significant difference between the Shannon index and the Simpson index of the CK and LI groups, but the Shannon index of fungi in the HI group was significantly higher than that of the CK and LI groups, and the Simpson index was significantly lower than that of the CK and LI groups. After tuber infestation, the high infestation rate group's mycorrhizosphere soil fungal richness and diversity were significantly higher than those of the low infestation and CK groups.

**Table 3.** Mycorrhizosphere soil fungal richness and diversity indices for different levels of truffle infestation.

| Fungal Samples | Species Richness | Species Diversity | |
|---|---|---|---|
| Classification | OTUs Observed | Shannon | Simpson |
| CK | 275.00 ± 21.52 a | 2.19 ± 0.05 b | 0.2384 ± 0.01711 b |
| LI | 308.33 ± 35.13 a | 2.37 ± 0.13 b | 0.2023 ± 0.02885 b |
| HI | 256.33 ± 20.03 a | 2.59 ± 0.96 a | 0.1517 ± 0.01504 a |
| *p*-value | 0.127 | 0.008 | 0.010 |
| F-value | 2.976 | 12.014 | 10.937 |

Note: Fungal diversity index based on OTU levels. Values are mean ± standard deviation (*n* = 3). Values followed by different lowercase letters indicate significant differences (*p* < 0.05) among samples within a line. CK is the uninoculated control group, LI is the low-infestation group, and HI is the high-infestation groupings.

Fungal OTUs belonged to 9 phylums, 30 classes, 61 orders, 128 families, 224 genera and 315 species. According to PCoA analysis (Figure 3), the different infestation groups had significantly distinct fungal community structures (ANOSIM, R = 0.852, *p* = 0.001). Following the infestation of pecan seedlings by truffles, the mycorrhizosphere soil's microbial community structure underwent significant alteration, with the highly infested group, the lowly infested group, and the uninoculated group being separated into three different

species, demonstrating great specificity between groups. From the histogram of fungal community composition (Figure 4), it can be found that the abundance of *Chaetomiaceae* in the three groups of samples was very high, especially in the control group, which accounted for more than 60% of the total, and the abundance in the high-infestation group and the low-infestation group was a little lower, which was around 40% and 20%, respectively. Apart from *Chaetomiaceae* fungi, members of the *Pyronemataceae* family and certain members of the *Sordariales* order also have a high abundance. Furthermore, among the three sets of samples, a number of unclassified fungus made up a sizeable fraction, with their share topping 40% in the LI group. The Wayne diagram (Figure 5) showed that the HI group enjoyed 64 specific fungal OTUs, the LI group 118, and the CK group 77, for a total of 184 OTUs among the three groups. The differential marker fungi in the high-infestation group, according to the LEfSe analysis results (Figure 6) (LDA > 4, *p* < 0.05), were *Rozellomycota* and *lasiosphaeriaceae*. *Sphaerosporella* and an unclassified fungus were among the marker flora in the low-infestation group, while *Chaetomium* and *Gilmaniella* were mostly found in the phylum *Ascomycetes* in the marker flora of the control group.

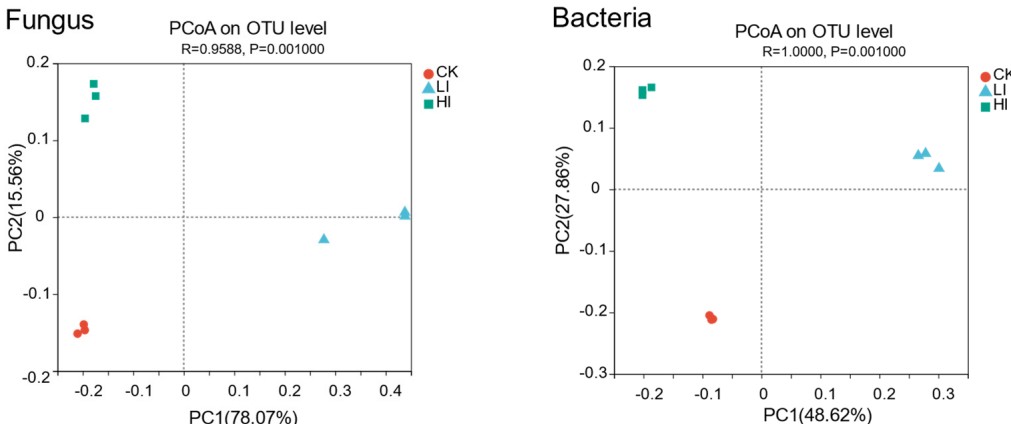

**Figure 3.** PCoA plots of soil fungal and bacterial communities of samples with different levels of infestation. Fungal communities are shown on the left and bacterial communities on the right. CK is the uninoculated control, LI is the low-infestation group, and HI is the high-infestation group. Note: *X*-axis and *Y*-axis represent the two selected principal coordinate axes, and the percentage indicates the value of the degree of explanation of the differences in the composition of the samples by the principal coordinate axes; the scales of *X*-axis and *Y*-axis are relative distances without practical significance; the points of different colors or shapes represent samples of different groupings, and the closer the points of the two samples are to each other, it indicates the more similar the composition of the two sample species is.

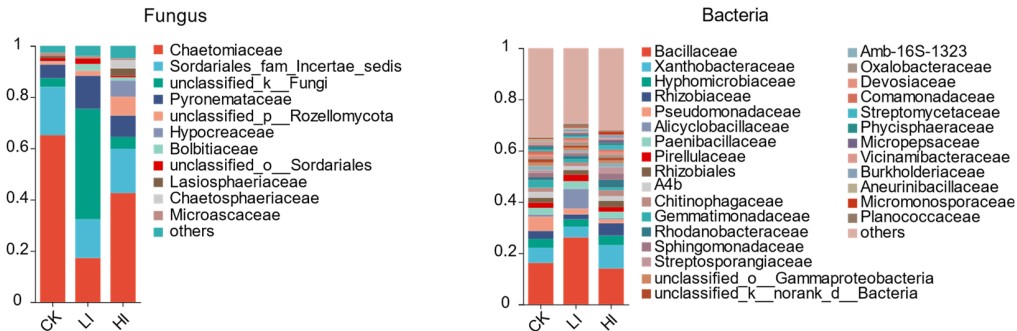

**Figure 4.** Family-level taxonomic composition of soil fungal-bacterial communities in Mycorrhizosphere pecan in different groups. CK is the uninoculated control group, LI is the low-infestation group, and HI is the high-infestation groupings.

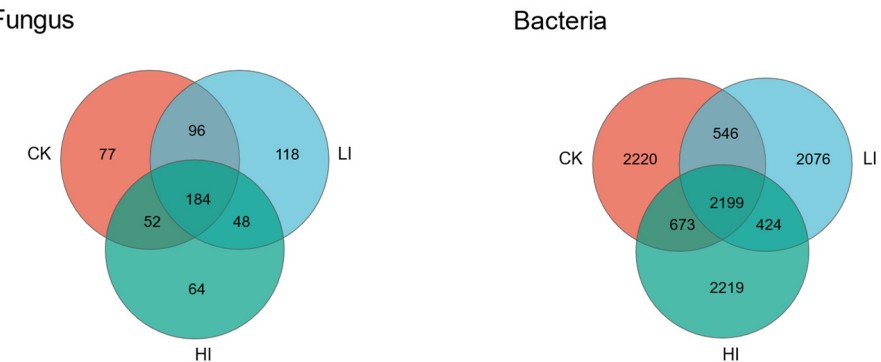

**Figure 5.** Venn diagrams of the unique and shared operational taxonomic units (OTUs) of soil fungal-bacterial communities in Mycorrhizosphere in different groupings. CK is the uninoculated control group, LI is the low-infestation group, and HI is the high-infestation groupings.

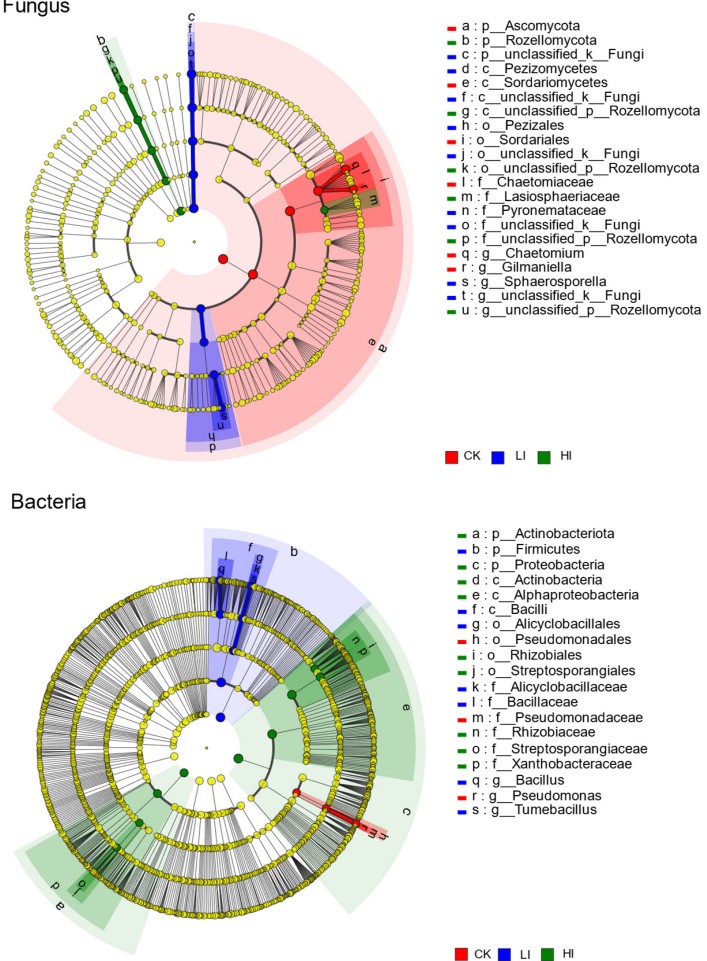

**Figure 6.** Tree diagram of LEfSe multilevel species analysis of rhizosphere soil of pecan in different groupings. linear discriminant analysis (LDA) was performed on pecan samples under different grouping conditions based on taxonomic composition to identify fungi and bacteria that had a significantly different effect on sample circling. The classification level of the tree diagram is from phylum to genus. Node size indicates the relative abundance of each sequence. Nodes of different colors indicate microorganisms that are significantly enriched in the corresponding group and have a significant impact on inter-group differences. Yellow nodes indicate microorganisms that did not differ significantly in any of the different groups. CK is the uninoculated control group, LI is the low-infestation group, and HI is the high-infestation group.

The 16S sequencing results obtained a total of 618,155 sequences, which were classified into 10,357 bacterial OTUs, and as shown in Table 4, the number of OTUs in the three treatment groups averaged between 2965 and 3169, which did not result in a significant difference, but unlike the fungi, in terms of $\alpha$-diversity, the bacterial diversity in the three groups of samples was very low, and the Shannon indexes in the CK group and the HI group were The Shannon index was significantly higher in the CK and HI groups than in the LI group. The Simpson index was significantly higher in the LI group than in the CK and HI groups, whereas there was no statistically significant difference in bacterial abundance and diversity between the CK and HI groups.

**Table 4.** Mycorrhizosphere soil bacteria richness and diversity indices for different levels of truffle infestation.

| Bacteria Samples | Species Richness | Species Diversity | |
|---|---|---|---|
| Classification | OTUs Observed | Shannon | Simpson |
| CK | 3169.00 $\pm$ 17.69 a | 6.62 $\pm$ 0.04 a | 0.0064 $\pm$ 0.00057 b |
| LI | 2965.00 $\pm$ 305.50 a | 6.15 $\pm$ 0.18 b | 0.01669 $\pm$ 0.00280 a |
| HI | 3069.67 $\pm$ 140.32 a | 6.15 $\pm$ 0.18 b | 0.00554 $\pm$ 0.00007 b |
| *p*-value | 0.462 | 0.003 | <0.0001 |
| F-value | 0.826 | 17.175 | 42.500 |

Note: Bacteria diversity index based on OTU levels. Values are mean $\pm$ standard deviation (*n* = 3). Values followed by different lowercase letters indicate significant differences (*p* < 0.05) among samples within a line. CK is the uninoculated control group, LI is the low-infestation group, and HI is the high-infestation groupings.

Bacterial OTUs belonged to 35 phylums, 116 classes, 296 orders, 476 families, 914 genera and 2226 species. *Bacillaceae, Xanthobacteriaceae, Hyphomicrobiacae,* and *Rhizobiaceae* occupied the top four positions in terms of bacterial community abundance, according to the histogram of bacterial community composition (Figure 4), while the proportion of *Bacillaceae*, the most abundant bacterial community in the sample community, was only 20%. The results of PCoA (Figure 3) showed that there were also significant differences between the different subgroups of the bacterial communities (ANOSIM, R = 1.000, *p* = 0.001). Significant variations in the bacterial community's makeup in the soil accompanied the establishment of pecan–truffle mycorrhizae and various degrees of infestation. The HI group had 2219 unique bacteria OTUs, the LI group 2076, and the CK group 2220, for a total of 2199 OTUs in all three groups, according to the Wayne diagram of bacterial species (Figure 5). According to the results of the LEfSe multilevel species difference discriminant analysis (Figure 6; LDA > 4, *p* < 0.05), the marker species for the low-infestation group were Bacillus and *Tumebacillus*, while those for the high-infestation group were *Rhizobiaceae, Xanthobacte-riaceae,* and *Streptococcaceae. Pseudomonas* was the bacterial marker species in the control group.

### 3.5. Association of Mycorrhizosphere Soil Microbial Community Composition with Environmental Factors

The RDA1 axis accounted for 5.97% of the variance in the fungal community composition, while the RDA2 axis accounted for 77.45% of the variance in the redundancy analysis of the association between fungal community composition and environmental parameters (Figure 6). $NH_4^+$-N and $Ca^{2+}$ content in the mycorrhizosphere soil had no significant effect on the fungal community composition. Fungal community variation in the highly infested group was significantly positively correlated with $NO_3^-$-N content and significantly negatively correlated with PH, AK, and AP content, while the exact opposite was true for the uninoculated control group, which was significantly positively correlated with PH, AK, and AP content and significantly negatively correlated with nitrate nitrogen content. The correlation heatmap showed (Figure 7, left) that nitrate nitrogen content was significantly positively correlated (*p* < 0.05) with the relative abundance of some of the top 20 fungi in the fungal community, one of which was identified at the genus level, two of

which were identified at the family level, and one of which was identified at the phylum level, namely *Collarina*, *Chaetosphaeriaceae*, *Sordariaceae*, and *Rozellomycota*; the levels of PH, AK, and AP were significantly positively correlated with the relative abundance of *Gilmaniella* and *Cephalotrichum*.

The RDA1 axis accounted for 9.86% and the RDA2 axis for 84.57% of the variation in bacterial community composition in the redundancy analysis of the association between bacterial community composition and environmental conditions (Figure 7, right). The analysis's findings also revealed that the quantity of fast-acting potassium and exchanged calcium ions in the soil's mycorrhizosphere had no significant impact on the diversity of the bacterial population. In contrast to the unvaccinated control group, where bacterial community variation was significantly positively correlated with PH and effective phosphorus content and negatively correlated with $NH_4^+$-N and $NO_3^-$-N content, the highly infested group's bacterial community variation was significantly positively correlated with ammonium and nitrate nitrogen content. According to the correlation heat map, the relative abundance of *Bauldia* was significantly and positively connected with the contents of $NH_4^+$-N and $NO_3^-$-N, whereas *Pseudomonas* and *Gemmatimonas* were considerably and significantly correlated with the contents of PH and AP.

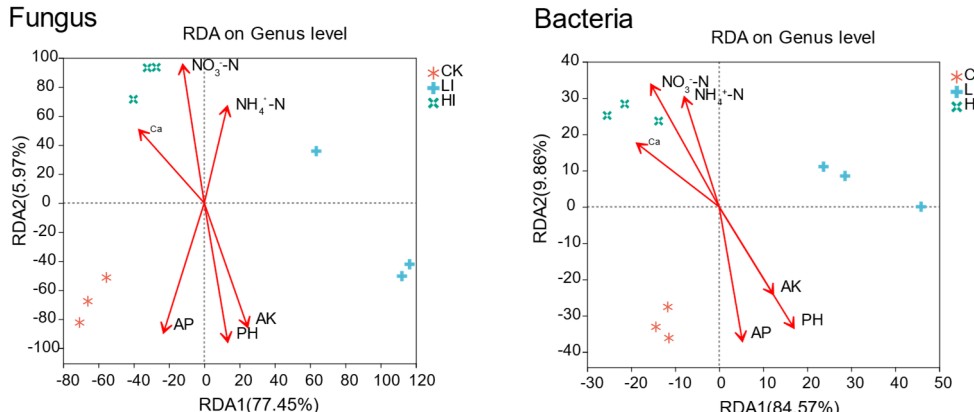

**Figure 7.** RDA plots of pecan soil samples in different groupings. Different colors or shapes of points in the graph indicate different sample groups under different environments or conditions; the red arrows indicate quantitative environmental factors; the length of the environmental factor arrows can represent the degree of influence of the environmental factor on the species data (the amount of explanation); the angle between the arrows of the environmental factor represents the positive and negative correlation (acute angle: positive correlation; obtuse angle: negative correlation; right angle: no correlation); the arrow of the environmental factor is projected from the sample point to the quantitative environmental factors, and the distance of the projection point from the origin represents the relative influence of the environmental factor on the distribution of the sample communities. The projections were made from the sample points to the arrows of the environmental factors, and the distances of the projected points from the origin represented the relative influence of the environmental factors on the distribution of the sample communities. Fungal community on the left and bacterial community on the right. CK is the uninoculated control group, LI is the low-infestation group, and HI is the high-infestation group.

## 4. Discussion

Microbial communities are dynamically shaped by various factors, environmental factors including soil, climate, region, and time of day [37], as well as characteristics of the host plant itself, including its species and developmental stage, play significant roles [38]. Previous studies have shown that microorganisms in the mycorrhizosphere soils of the same species may also exhibit different characteristics due to infestation by other organisms [39]. Following truffle inoculation, the Shannon index of soil fungi in the samples of the high-infestation group was significantly higher than that of the low-

infestation group and the control group, and the Simpson index was significantly lower ($p < 0.05$). This shows that the truffle infection increased the fungal diversity of the host plant's mycorrhizosphere soil and caused its microbial community structure to evolve towards a healthier, more stable, and beneficial aspect (Table 3). In comparison to fungi, there were more bacteria in the sample soils; however, there was no significant correlation between the inoculation of truffles and the diversity index of the bacterial community as a whole, with the HI group being comparable to the CK group in terms of bacterial $\alpha$-diversity, and the LI group being lower than the remaining two groups. The effects of inoculation seem to be better reflected in specific microbial species; the HI group exhibited 64 specifics fungal OTUs and 2219 specific bacterial OTUs, according to an examination of the species Wayne diagram (Figure 5), and these specialized microorganisms are likely to be closely related to mycorrhizal formation in pecan–truffle. The three groups shared 184 fungal OTUs and 2199 bacterial OTUs, and these shared microorganisms may constitute the core microbial community of the pecan species [40]. The microbial communities of the three groups of samples varied significantly, and after PCoA decentralized analysis, the fungal and bacterial data of the samples formed three separate clusters. *Rozellomycota* and *lasiosphaeriaceae* were discovered to be the marker fungi of the differences in the high-infestation group using LEfSe multilevel species difference discriminant analysis (Figure 6), whereas the marker flora of the low-infestation group consisted of an unclassified fungus and *sphaerosporella*; these fungal community differences can be explained by competition, with species interactions and competition often determining root colonization [41,42]. The marker fungi of the high-infestation group may not be in the same ecological niche as the truffles or have a supportive role, and thus can coexist well, while the fungi of the low-infestation group may be in ecological niche competition with the truffles and eventually dominate the soil ecosystem. The *Rhizobiaceae*, *Xanthobacteriaceae*, and *Streptococcaceae* bacterial species are bacterial differential marker species in the high infestation category. Previous research also revealed that these bacteria typically formed dominant genera in the tuberous mycorrhizal soils of the Huashan pine [43] (*Pinus armandii*), European hazelnut [44,45] (*Corylus avellana*), and European goosefoot [46] (*Carpinus betulus*). They are recruited by mycorrhizal seedlings in a variety of habitats with various soils, climatic conditions, and host plants, and they probably contribute significantly to the growth in and development of truffles. *Bacillus* and *Tu-mebacillus* were the bacterial marker groups in the low-infestation group. *Pseudomonas* bacteria were the bacterial marker groups in the control group. These findings imply that mycorrhizosphere effects [47] are still significantly present during infestation of pecan by truffles, and that fungi selectively shape the associated microbial communities by altering root secretions, leading to quantitative and qualitative changes in root secretions between the mycorrhizosphere and mycorrhizal intervals [48]. These microorganisms tend to serve the needs of both the host plant and the mycorrhizal fungi. However, they are far from being well understood, and their roles in the plant–mycorrhizal and fungus–bacteria interaction network, as well as their mechanisms of action, remain to be elucidated [49]. The RDA1 and RDA2 axes explained a total of 83.42% of the variation in the composition of the flora in the redundancy analysis of the association between the composition of the fungal community and environmental factors, with AP ($p = 0.011$), PH ($p = 0.022$), and $NO_3^-$-N ($p = 0.024$) being the most significant drivers of variation in the fungal flora (Figure 7, left). The *Collarina* genus and the *Rozellomycota*, among other fungus, were substantially connected with these drivers, according to the heat map of the correlation (Figure 8, left). The RDA1 and RDA2 axes explained a total of 94.43% of the variation in the composition of the bacterial community in the redundancy analysis of the association between bacterial community composition and environmental factors (Figure 7, right). The most significant drivers of bacterial community changes were AP ($p = 0.001$), PH ($p = 0.006$), $NO_3^-$-N ($p = 0.012$), and $NH_4^+$-N ($p = 0.026$), and correlation heatmaps (Figure 8, right) revealed that bacteria of the genera *Pseudomonas* and *Gemmatimonas* were strongly associated with these environmental factors.

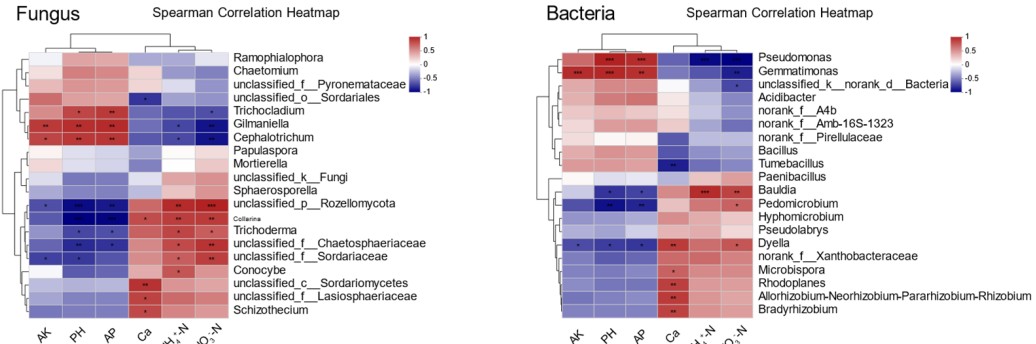

**Figure 8.** Heat map of correlation between relative abundance of dominant soil microorganisms and soil chemical properties. the horizontal and vertical axes indicate the top 20 microbial species in terms of soil chemical properties and relative abundance, respectively. *, **, *** denote $p < 0.05$, $p < 0.01$, and $p < 0.001$, respectively. AP, AK, and Ca are effective phosphorus, effective potassium, and exchangeable calcium contents, respectively; $NO_3^--N$ and $NH_4^+-N$ are nitrate nitrogen and ammonium nitrogen contents.

Most host plants are thought to benefit from mycorrhizal connections in terms of growth and development because they have better access to nutrients and can withstand biotic and abiotic stresses [50]. However, the quality of pecan seedlings inoculated with ectomycorrhizal mycorrhizae in the present trial was not significantly different from that of the blank control group (Table 1), which may be due to the fact that while the seedlings in the trial were containerized, container blockage affects the construction of mycelial networks below the epiphytic mycelium ground floor [51]. On the other hand, in the early stages of development, both seedlings and EMF require a lot of nutrients for growth and development, and EMF's nutritional depletion may render this positive feedback insignificant [25]. The root system of seedlings in the highly infested group in the experiment had significantly higher POD activity than that of the other two groups. POD activity is closely related to the plant's defense mechanism against the disease. Studies show that soybean that had been inoculated with AMF had significantly higher POD activity in the face of the disease. In the face of disease infestation, it was noticeably greater and had a lower incidence [52]. Therefore, inoculation with truffles may help to improve the resistance of pecan to certain diseases.

With regard to the physicochemical properties of the mycorrhizosphere soil, our study showed that the soil PH of seedlings inoculated with truffles was significantly lower than that of the uninoculated group, and this trend became more and more obvious as the infestation rate increased (Table 2). The organic acids produced during mycorrhizal development may be responsible for this [53]. Additionally, it has been demonstrated that the soil created by truffles is greatly enriched with acidic chitinase, which is secreted by specific actinomycetes that are widely distributed in mycorrhizal soils [54]. In our research, $NO_3^--N$ levels in the mycorrhizosphere soil of seedlings in the highly infected group were significantly higher than those in the blank control group and the low infestation group (Table 2). It has been demonstrated that ectomycorrhizal mycorrhizae have a greater preference for organic nitrogen and very limited uptake of $NO_3^--N$, and only a few $NO_3^-$-transporters have been identified in ECM fungi [55], which may be a key factor contributing to the increase in $NO_3^--N$ content in the highly infested group. Our results showed that the samples from the highly infested group had significantly lower AP and AK contents than those from the control group. AP and AK can be absorbed and utilized by plants and microorganisms [56]. Due to the limited nutrient supply of the soil in the containers, the mycorrhizosphere microorganisms of the pecan seedlings inoculated with truffle fungi may have a higher need for these nutrients in this study [57]. This could play a significant role in explaining why the effective potassium and phosphorus contents were lower in the experimental group than in the control group. In this study,

there was no significant difference in the change in exchange state calcium content in the mycorrhizosphere soil of the three groups, which may be due to the fact that the tuber inoculation did not have a significant effect on the uptake of exchange state calcium in the test seedlings.

## 5. Conclusions

The richness and diversity of the mycorrhizosphere soil fungal community of pecan seedlings was increased after tuber inoculation. Additionally, tuber inoculation changed the microbial community's composition in the pecan seedlings' mycorrhizosphere by recruiting different fungi, such as *Rozellomycota* and *Lasiosphaeriaceae*, as well as different bacteria, such as *Rhizobiaceae*, *Xanthobacteriaceae*, and *Streptococcaceae*. These microorganisms caused a drop in the PH and AP level of the soil as well as a rise in the $NO_3^-$-N content, thus shaping a soil environment suitable for the growth of truffles. This study reveals the microbial mechanisms that affect the formation of pecan-truffle mycorrhizal symbiosis, provides new ideas and insights for people to improve the rate of truffle infestation in the cultivation and production practice of pecan–truffle mycorrhizal seedlings.

**Supplementary Materials:** The following supporting information can be downloaded at: https://www.mdpi.com/article/10.3390/f14102078/s1, Table S1: Statistical table of mycorrhizal infestation rate of truffles.

**Author Contributions:** Author Contributions: Experimental design, H.C., J.L. and F.P.; sample collection, H.C. and J.W.; data analysis and writing—original draft preparation, H.C.; writing—review and editing, J.L., K.Z. and P.T.; project administration and funding acquisition, F.P. All authors have read and agreed to the published version of the manuscript.

**Funding:** The research was supported by the Technology Promotion and Demonstration Fund Project of China (Su [2022] TG04) and the National Key R&D Program Project, China (2021YFD1000403).

**Data Availability Statement:** Due to confidentiality agreements in the laboratory of the research group, the data may be used for other purposes or for other analytical work.

**Conflicts of Interest:** The authors declare no conflict of interest.

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
