# Peer review of "Effects of Truffle Inoculation on Root Physiology and Mycorrhizosphere Microbial Communities of Carya illinoinensis Seedlings"

_forests, doi:10.3390/f14102078_

Round 1

Reviewer 1 Report

The article is devoted to a very interesting and relevant topic: the effect of truffle inoculation on the physiology of the roots and microbial communities of the mycorrhizosphere of Carya illinoinensis seedlings.

What type of soil and where did you use for the study?

Formulas on lines 187, 251 should be numbered and concise with abbreviations indicated under the formula.

Reviewer 2 Report

The manuscript entitled: „Effects of truffle inoculation on root physiology and mycorrhizosphere microbial communities of Carya illinoinensis seedlings” presents interesting study on effects of  truffle inoculation.

The manuscript contains some drawbacks:

1) Please follow the guidelines for authors, e.g. words in the title should start with capital letters.

2) Aim of the study presented in the end of Introduction should be more specific. Current aim contains information what was done and what results were obtained but there is lack information about the aim.

3) Calculation of infestation rate was based on small sample size. For such sample size confidence interval for proportion is very wide. It should be presented in the results that confidence of the result is very low. Please add range of confidence interval to the Table S2 as an additional column.

4) Tables and figures should be self-explanatory, i.e. clear enough without reading all the manuscript. Some of them are not sufficiently described. For example it is not clear what is presented in Table 1. It is mean and SD? Please add more specific captions. Moreover, I suggest to place letters which indicate homogenous groups next to means because means are compared not SDs.

5) Lines 255-256: There is information: “Duncan's Multiple Polar Difference Test, and Least Significant difference (LSD)”. Which of these two methods of multiple comparisons was used for which analysis, for example it is unknown what method (Duncan’s or LSD) was applied for comparisons presented in Table 1, 2 and 3.

6) Lack of information about RDA in Material and methods while the results of RDA are presented in Results section.

7) Typo in line 583: “Conflflicts”; Line 472: “p<05” (it should be p<0.05?). Please be more careful.

Round 2

Reviewer 2 Report

The manuscript was substantially improved according most of my comments. I still have some comments. The authors have not calculated the confidence intervals for proportions presented in supplementary material. It can be performed using various methods: https://en.wikipedia.org/wiki/Binomial_proportion_confidence_interval

If it is possible, please calculated CI for these proportions.

Please use italic for latin name of the species, eg. line 698 and 713.
